# Friendship paradox biases perceptions in directed networks

Nazanin Alipourfard[1,3]*, Buddhika Nettasinghe[2,3]*, Andrés Abeliuk[1], Vikram Krishnamurthy[2] & Kristina Lerman[1]

Social networks shape perceptions by exposing people to the actions and opinions of their peers. However, the perceived popularity of a trait or an opinion may be very different from its actual popularity. We attribute this perception bias to friendship paradox and identify conditions under which it appears. We validate the findings empirically using Twitter data. Within posts made by users in our sample, we identify topics that appear more often within users' social feeds than they do globally among all posts. We also present a polling algorithm that leverages the friendship paradox to obtain a statistically efficient estimate of a topic's global prevalence from biased individual perceptions. We characterize the polling estimate and validate it through synthetic polling experiments on Twitter data. Our paper elucidates the non-intuitive ways in which the structure of directed networks can distort perceptions and presents approaches to mitigate this bias.

[1] Information Sciences Institute, 4676 Admiralty Way, Marina Del Rey, Los Angeles, CA 90292, USA. [2] Cornell University, 418 Phillips Hall, Ithaca, NY 14853, USA. [3] These authors contributed equally: Nazanin Alipourfard, Buddhika Nettasinghe. *email: nazanina@isi.edu; dwn26@cornell.edu

**W**e observe our peers to learn social norms, assess risk, or copy behaviors. However, these observations can be systematically biased[1–7], distorting how we see the world. One of the better known sources of bias is the friendship paradox in social networks[8], which states that people are less popular than their friends are, on average. Consequences of friendship paradox can skew how we compare ourselves to friends: people tend to be less happy than their friends are[9], and researchers tend to have less impact than their co-authors do[10], on average. In fact, any trait correlated with popularity is likely to be misperceived[11,12]. This may explain why adolescents systematically overestimate how much their peers drink or engage in risky behaviors[2,5] and why social media use is often associated with negative social comparisons[13].

In contrast to friendships, many online social networks are directed. On Twitter, for example, we subscribe to, or follow, others to see their posts, but the information does not flow in the opposite direction, unless those people also follow us back. For convenience, we refer to people whose posts we see in our social feeds as our friends, and those who see our posts as followers. Clearly, this nomenclature does not imply a bidirectional friendship relationship. An individual's in-degree is the number of his or her friends, and the out-degree is the number of followers. The asymmetric nature of links in directed networks leads to four variants of the friendship paradox[14]: your friends (or followers) have more friends (or followers) than you do, on average. Empirically, this effect can be quite large, with upwards of 90% of social media users observing that they have a lower in-degree and out-degree than both their friends and followers[15]. However, the conditions under which these four variants of the paradox exist have not been comprehensively analyzed. We carry out the analysis to show that while two variants of the friendship paradox occur in any directed network[16], the remaining two exist only if an individual's in-degree and out-degree are correlated.

Friendship paradox can systematically skew individual's observations of the network's state. We consider directed networks where nodes have a trait, such as gender, political affiliation, or whether they used a certain hashtag in their posts. The trait's global prevalence is simply the fraction of all nodes with that trait. On the other hand, its observed prevalence is the fraction of friends that have the trait. In networks where the more influential (higher out-degree) nodes are likely to have the trait, its observed prevalence will be substantially higher than its actual prevalence. Our analysis shows that, similar to the generalized friendship paradox in undirected networks[12,17], correlation between nodes' trait and their out-degree amplifies this perception bias.

In reality, an individual's perception of a trait is shaped by its local prevalence among his or her friends. In this paper, we identify a new paradox in directed networks, as a result of which a trait will appear to be significantly more popular locally among an individual's friends, than it is globally among all people. We show that this effect is stronger in networks where higher out-degree nodes with the trait are connected to nodes with a lower in-degree.

Surprisingly, although individual observations are biased, we can still robustly estimate the global prevalence of the trait. We present a polling algorithm that obtains a statistically efficient estimate of a trait's global prevalence, with a smaller error than alternative polling methods. Proposed method leverages friendship paradox to reduce the error of the polling estimate by trading off the bias of the estimate and its variance. We analytically characterize this trade-off and provide an upper bound for the variance.

We also show that perception bias can be large in a real-world network. To this end, we extracted a subgraph of the directed Twitter social network and collected messages posted by users within this subgraph. Treating the occurrence of particular hashtags within messages as traits or topics enables us to measure the perception bias. We identify hashtags that appear much more frequently within users' social feeds than they do among all messages posted by everyone, leading users to overestimate their prevalence. We also validate the performance of the proposed polling algorithm through synthetic polling experiments on the Twitter subgraph.

This paper elucidates some of the non-intuitive ways that directed social networks can bias individual perceptions. Since collective phenomena in networks, such as social contagion and adoption of social norms, are driven by individual perceptions, the structure of networks and the paradoxes endemic in them can impact social dynamics in unexpected ways. Our work shows how we can begin to quantify and mitigate these biases.

## Results

**Basic concepts and definitions**. Consider a directed network $G = (V, E)$, with $\{V\}$ nodes and $\{E\}$ links. A link $(i, j)$ pointing from $i$ to $j$ indicates that $i$ is a friend of $j$ or equivalently, $j$ follows $i$. Here, the direction of the link indicates the flow of information. The out-degree of a node $v$, $d_o(v)$, measures the number of followers it has, and its in-degree, $d_i(v)$, the number of friends.

We define three random variables, $X$, $Y$, and $Z$ that correspond to different node sampling methods. A node $v$ with out-degree $d_o(v)$ has that many followers, or equivalently, $v$ is a friend to $d_o(v)$ number of nodes. Therefore, a node $Y$ that is obtained from $V$ by sampling proportional to out-degree of nodes is called a random friend. Similarly, a node $v$ that has $d_i(v)$ links pointing to it is a follower of $d_i(v)$ other nodes. Therefore, a node $Z$ that is obtained from $V$ by sampling proportional to in-degree of nodes is called a random follower. Below, we formalize these terms.

1. Random node $X$ is a uniformly sampled node from $V$:

$$\mathbb{P}(X = v) = \frac{1}{N} \quad \forall v \in V. \tag{1}$$

2. Random friend $Y$ is a node sampled from $V$ proportional to its out-degree:

$$\mathbb{P}(Y = v) = \frac{d_o(v)}{\sum_{v' \in V} d_o(v')}, \quad \forall v \in V. \tag{2}$$

3. Random follower $Z$ is a node sampled from $V$ proportional to its in-degree:

$$\mathbb{P}(Z = v) = \frac{d_i(v)}{\sum_{v' \in V} d_i(v')}, \quad \forall v \in V. \tag{3}$$

For any directed network, the average in-degree $\mathbb{E}\{d_i(X)\} = \frac{\sum_{v \in V} d_i(v)}{N}$ and the average out-degree $\mathbb{E}\{d_o(X)\} = \frac{\sum_{v \in V} d_o(v)}{N}$ are the same. Here $\mathbb{E}$ denotes the expectation operator. Therefore, we use $\overline{d}$ to denote both average in-degree and average out-degree of a random node $X$: $\overline{d} = \mathbb{E}\{d_o(X)\} = \mathbb{E}\{d_i(X)\}$.

**Friendship paradox in directed networks**. Four different variants of the friendship paradox exist in directed networks[14]. The first two, state that (1) random friends have more followers than random nodes do, and (2) random followers have more friends than random nodes do (on average). The magnitudes of these are set by the variance of the in-degree and out-degree distributions of the underlying network. Mathematically, these two friendship paradoxes can be stated as

- Random friend $Y$ has more followers than a random node $X$, on average:

$$\mathbb{E}\{d_o(Y)\} - \overline{d} = \frac{\operatorname{Var}\{d_o(X)\}}{\overline{d}} \geq 0. \qquad (4)$$

- Random follower $Z$ has more friends than a random node $X$, on average:

$$\mathbb{E}\{d_i(Z)\} - \overline{d} = \frac{\operatorname{Var}\{d_i(X)\}}{\overline{d}} \geq 0. \qquad (5)$$

For the derivation, please see Supplementary Note 1.

The remaining two variants of the friendship paradox state that (3) random friends have more friends than random nodes do, and (4) random followers have more followers than random nodes do (on average). In contrast to the first two variants of the paradox stated above, the remaining two variants require positive correlation between the in-degree and the out-degree of nodes in the network:

- Random friend $Y$ has more friends than a random node $X$, on average:

$$\mathbb{E}\{d_i(Y)\} - \overline{d} = \frac{\operatorname{Cov}\{d_i(X), d_o(X)\}}{\overline{d}} \geq 0. \qquad (6)$$

- Random follower $Z$ has more followers than a random node $X$, on average:

$$\mathbb{E}\{d_o(Z)\} - \overline{d} = \frac{\operatorname{Cov}\{d_i(X), d_o(X)\}}{\overline{d}} \geq 0. \qquad (7)$$

For the derivation, please see Supplementary Note 1.

Equations (6) and (7) state that in networks where the in- and out-degrees of a random node are positively correlated, (1) the expected number of friends of a random friend is greater than the expected number of friends of a random node, and (2) the expected number of followers of a random follower is greater than that of a random node. The mathematical formulations of the friendship paradox in directed networks were independently proved recently in ref. [16] utilizing vector norms.

To give additional intuition, Fig. 1 illustrates the above four variants of the friendship paradox in the subgraph of the Twitter social network (see the "Methods" secion), showing the fraction of individuals with a specific in-degree (or out-degree) who experience the paradox. Note that this fraction is high: at least half of the users with 100 or fewer friends (or followers) observe that they are less popular and well-connected than their friends and followers are on average. The noise in Fig. 1 likely stems from Twitter's follow limits. When individuals reach the limit, they must curate their social links more deliberately and recruit more followers before they can add more friends.

**Global perception bias**. When nodes have distinguishing traits or attributes, the friendship paradox can bias perceptions of those attributes. For simplicity, we assume that each node has a binary-valued attribute ($f: V \rightarrow \{0, 1\}$). Such binary functions are useful for representing, among others, voting preferences (Democratic or Republican), demographic characteristics (female or male), contagions (infected vs. susceptible), or the spread of information in networks (using a particular hashtag or not).

The global prevalence of the attribute in a directed network is given by $\mathbb{E}\{f(X)\}$, the expected value of the attribute of a random node $X$. In other words, when only 5% of nodes have the attribute $f(v) = 1$, its expected value is $\mathbb{E}\{f(X)\} = 0.05$.

Nodes' perceptions of the prevalence of the attribute, however, are determined by its value among their friends, i.e., $\mathbb{E}\{f(Y)\}$, the expected attribute value of a randomly chosen friend $Y$. On Twitter, this translates into how many people see the topic in their social feed, since the feed aggregates posts made by friends. Under some conditions, the perceived prevalence of the attribute $\mathbb{E}\{f(Y)\}$ will be very different from its actual prevalence $\mathbb{E}\{f(X)\}$. We define this as global perception bias:

$$B_{\text{global}} = \mathbb{E}\{f(Y)\} - \mathbb{E}\{f(X)\} = \frac{\operatorname{Cov}(f(X), d_o(X))}{\overline{d}} = \frac{\rho_{d_o,f}\sigma_{d_o}\sigma_f}{\overline{d}}, \qquad (8)$$

where $\rho_{d_o,f}$ is the Pearson correlation coefficient between out-degree and attribute value of a random node, $\sigma_{d_o}$ is the standard deviation of the out-degree distribution, and $\sigma_f$ is the standard deviation of the binary attributes (see Supplementary Note 1 for the derivation).

When the attribute is correlated with the out-degree ($\rho_{d_o,f} > 0$), a random friend's attribute is larger than the attribute value of a random node, on average. In undirected networks this effect is known as generalized friendship paradox[12], and it has the same intuition: when popular people (with many followers) are more likely to possess some trait ($\rho_{d_o,f} > 0$), that trait will be overrepresented among the friends of any individual. As a result, people will tend to overestimate the trait's prevalence. This may explain the observation that adolescents overestimate the number of smokers or heavy drinkers among their peers[2]. All that is required for the bias to hold is for peers engaging in risky behaviors to tend to be more popular.

Note that the magnitude of the friendship paradox

$$S_{\text{FP}} = \mathbb{E}\{d_o(Y)\} - \overline{d} = \frac{\sigma_{d_o}^2}{\overline{d}}$$

increases with the standard deviation of the out-degree distribution ($\sigma_{d_o}$) and decreases with the average degree ($\overline{d}$). Global perception bias $B_{\text{global}}$ also increases with $\sigma_{d_o}$ and decreases with $\overline{d}$ when the correlation coefficient $\rho_{d_o,f}$ remains fixed. Hence, friendship paradox amplifies global perception bias, increasing the deviation between the actual and observed prevalence of the attribute in the network.

Additional perception biases can arise in directed networks. Recall that a random friend $Y$ is an individual sampled with a probability proportional to the out-degree, and a random follower $Z$ is an individual sampled with a probability proportional the in-degree. The random friend $Y$ can be thought of as a person being observed, whereas a random follower $Z$ is a person who is observing. In this context, the perception bias $B_{\text{global}} = \mathbb{E}\{f(Y)\} - \mathbb{E}\{f(X)\}$ compares the opinion of a random person being observed with the global (true) prevalence. By the same token, the quantity $\mathbb{E}\{f(Z)\} - \mathbb{E}\{f(X)\}$ compares the opinion of a random observer with the global prevalence. The difference

$$\mathbb{E}\{f(Y)\} - \mathbb{E}\{f(Z)\} = \frac{1}{\overline{d}} \mathbb{E}\{f(X)(d_o(X) - d_i(X))\}$$

can then be thought of as the expected difference of the opinions between the observed and the observer pair chosen randomly from the network. This interpretation opens up a causal perspective of the perception bias in directed networks for future work.

**Local perception bias**. One problem with using $B_{\text{global}}$ (Eq. (8)) to measure perception bias is that $\mathbb{E}\{f(Y)\}$ captures the expected value of the attribute among the friends of all individuals, rather than the friends of a randomly chosen individual $X$. In order to

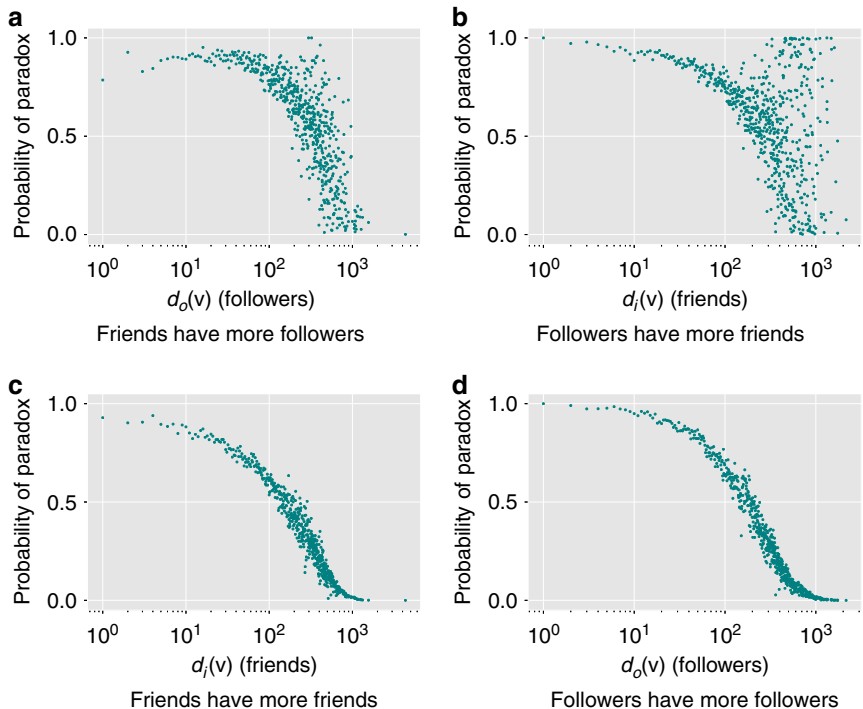

**Fig. 1 Illustration of the effects of the four versions of the friendship paradox using Twitter dataset described in the "Methods" section.** The sub-figures display the fraction of nodes (empirical probability of the paradox) of a particular degree whose **a** friends have more followers, **b** followers have more friends, **c** friends have more friends, and **d** followers have more followers, on average.

reflect more accurately how individual's perceptions are skewed by their friends, we propose a new measure of perception bias. To quantify this bias we begin by defining the perception $q_f(v)$ of an individual $v \in V$ about the prevalence of an attribute $f$ among his or her friends:

$$q_f(v) = \frac{\sum_{u \in \mathcal{F}(v)} f(u)}{d_i(v)}, \qquad (9)$$

where $\mathcal{F}(v)$ denotes the set of friends of $v$. We define local perception bias as the deviation of the expected perception of a trait of a random individual from its global prevalence:

$$B_{local} = \mathbb{E}\{q_f(X)\} - \mathbb{E}\{f(X)\}. \qquad (10)$$

To help understand $B_{local}$, we define the attention that a node $v \in V$ allocates to each of her friends:

$$\mathcal{A}(v) = \frac{1}{d_i(v)}.$$

The expression for attention is motivated by an observation that users with more friends tend to receive more messages[18], making them less likely to see any specific friend's post[19]. This allows us to succinctly express the expected perception of a random node $X$ as (see Supplementary Note 1 for the derivation)

$$\mathbb{E}\{q_f(X)\} = \overline{d} \cdot \mathbb{E}\{f(U)\mathcal{A}(V)|(U,V) \sim \mathrm{Uniform}(E)\}.$$

Here, $\overline{d}$ is the expected number of friends of a random node, and $U$ and $V$ denote the endpoints of a link sampled uniformly from $E$. Intuitively,

$$\mathbb{E}\{f(U)\mathcal{A}(V)|(U,V) \sim \mathrm{Uniform}(E)\}$$

represents the expected influence of an interaction along a link drawn at random from the network: i.e., the attribute $f(U)$ of the friend $U$ times the attention that the follower $V$ pays to that friend. Note that for simplicity we assumed that nodes divide their attention uniformly over all friends, though the analysis can

be extended to weighted networks, where weights model non-uniform attention, with individuals paying more attention to their more important or influential friends.

**Relationship between $B_{local}$ and $B_{global}$.** Local perception bias $B_{local}$ is a refinement of global perception bias $B_{global}$, which accounts for how individuals divide their attention in the network. Indeed, if the attention of followers is independent of the attribute of their friends, both measures are the same. Formally, $B_{global}$ and $B_{local}$ are equal if and only if the attribute $f(U)$ of $U$ and attention $\mathcal{A}(V)$ along a random link $(U,V)$ are uncorrelated, i.e.,

$$\mathrm{Cov}\{f(U), \mathcal{A}(V)|(U,V) \sim \mathrm{Uniform}(E)\} = 0, \qquad (11)$$

as we show in Supplementary Note 1.

On the other hand, positive local perception bias exists, i.e., $B_{local} \geq 0$, when the following conditions are met (see SI):

$$\mathrm{Cov}\{f(X), d_o(X)\} \geq 0 \quad \text{and,} \qquad (12)$$

$$\mathrm{Cov}\{f(U), \mathcal{A}(V)|(U,V) \sim \mathrm{Uniform}(E)\} \geq 0. \qquad (13)$$

The first condition (Eq. (12)) specifies positive correlation between the out-degree and the attribute of a random node, which occurs when popular nodes are more likely to have the attribute. This is a necessary and a sufficient condition for $B_{global} \geq 0$ (see Eq. (8)). The second condition (Eq. (13)) specifies positive correlation between the attention of a follower and the attribute of a friend, suggesting that nodes with an attribute are followed by nodes that divide their attention over few others. This is a necessary and a sufficient condition for $B_{local} \geq B_{global}$ (see Supplementary Note 1). Hence, these two conditions collectively are sufficient for positive local perception bias, leading individuals to overestimate the attribute's prevalence, i.e., $B_{local} \geq B_{global} \geq 0$. Analogously, changing the signs of Eqs. (12) and (13) leads to negative local perception bias (see Supplementary Note 2), which implies that nodes underestimate the prevalence of an attribute.

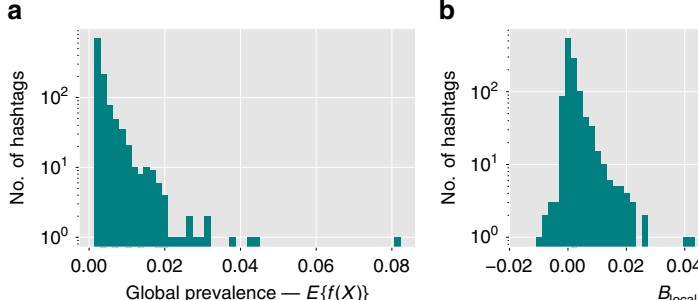

**Fig. 2 Global prevalence and local bias of popular hashtags.** Histogram of the distribution of **a** global prevalence $\mathbb{E}\{f(X)\}$ and **b** local perception bias $B_{\text{local}}$ of popular hashtags in the Twitter data. Local perception bias $B_{\text{local}}$ (overestimating the prevalence) exists for most hashtags.

Under other conditions $B_{\text{global}}$ and $B_{\text{local}}$ can differ significantly and even disagree, with one measure indicating that individuals are underestimating and the other indicating that they are overestimating the prevalence of an attribute. We analytically characterize the two cases where this occurs (see Supplementary Note 1):

1. Let $B_{\text{global}} < 0$, then $B_{\text{local}} > 0$ if and only if

$$\text{Cov}\{f(U), \mathcal{A}(V)|(U, V) \sim \text{Uniform}(E)\} > \frac{|B_{\text{global}}|}{\bar{d}}.$$

2. Let $B_{\text{global}} > 0$, then $B_{\text{local}} < 0$ if and only if

$$\text{Cov}\{f(U), \mathcal{A}(V)|(U, V) \sim \text{Uniform}(E)\} < \frac{-|B_{\text{global}}|}{\bar{d}}.$$

The first condition states that when $B_{\text{global}}$ is negative, $B_{\text{local}}$ can still be positive if sufficiently many nodes with an attribute have followers with high attention (because they divide it over few friends). Similarly, when $B_{\text{global}}$ is positive, $B_{\text{local}}$ can be negative when few of the nodes with an attribute have high attention followers, leading to a negative correlation between the attribute of a friend and the attention of his or her follower. The discrepancy exists because $B_{\text{global}}$ makes a mean-field approximation by assuming that the expected attribute value among friends of a random node $X$ is equal to the expected attribute value of a random friend $Y$ sampled from the entire network. In contrast, $B_{\text{local}}$ is a higher resolution measure that takes the underlying network structure into account via the correlation between the attribute of a friend and the attention of a follower.

*Relation to inversity in undirected networks*: Local perception bias is related to the concept of inversity[20], which is defined as the correlation coefficient of the two random variables $d(U)$ and $1/d(V)$ where, $d$ denotes the degree, and $(U, V)$ is a uniformly sampled link in an undirected network. Although the mathematical form of inversity is reminiscent of degree assortativity[21], it does not convey the same information. Kumar et al.[20] shows that the relation between global and local versions of the friendship paradox in undirected networks is characterized by inversity and not assortativity. Specifically, when inversity is positive, then the local version of the friendship paradox is larger in magnitude than the global version of the friendship paradox in undirected networks. This result can be obtained by extending our analysis to undirected networks and setting $f = d$ in the expressions for $B_{\text{global}}$ and $B_{\text{local}}$. In fact, Eq. (13) (which is a necessary and sufficient condition for $B_{\text{local}} \geq B_{\text{global}}$) in our paper generalizes their findings to directed networks and arbitrary exogenous attributes $f$.

**Empirical validation**. We used data from Twitter (see the "Methods" section) to compare the actual and perceived popularity of hashtags (i.e., topics) mentioned in text posts. We treat each hashtag $h$ as a binary attribute, with $f_h(v) = 1$ if a user $v$ used the hashtag $h$ in his or her posts.

Figure 2a displays the histogram of the prevalence ($\mathbb{E}\{f(X)\}$) of the 1153 most popular hashtags, each used by more than 1000 people in our data set. The bulk of these hashtags were used by fewer than 2% of the people, with the most popular hashtags being used by just 8% of the people in our sample. Figure 2b shows the histogram of local perception bias $B_{\text{local}}$ for all hashtags. Although its peak is at zero, the distribution is skewed, with 865 hashtags having a positive bias, meaning that they appear more popular than they really are. Measurements of individual's perception shows that most users in our sample overestimate how popular hashtags are (see Supplementary Note 2).

What hashtags are most biased? Figure 3 shows the top-20 and bottom-10 hashtags ranked by $B_{\text{local}}$ (see Supplementary Fig. 3 for the ranking of hashtags based on the global bias). Among the most positively biased hashtags are those associated with social movements (#ferguson, #mikebrown, #michaelbrown), memes and current events (#icebucketchallenge, #alsicebucketchallenge, #ebola, #netneutrality), sports and entertainment (#emmys, #robinwilliams, #sxsw, #applelive, #worldcup). For example, #ferguson, with $\mathbb{E}\{q_f(X)\} = 12.1\%$, is perceived as the most popular hashtag. While it is also one of the more widely used hashtags, with $\mathbb{E}\{f(X)\} = 3.1\%$, perception bias makes it appear about four times more popular to Twitter users than it actually is.

There are also negative biased hashtags, which appear less popular than they actually are. Among these hashtags are Twitter conventions aimed at getting more followers (#tfb, #followback, #follow, #teamfollowback) or more retweets (#shoutout, #pjnet, #retweet, #rt). Many of these hashtags are actually among the top-20 most popular Twitter hashtags (#oscars, #tcot, #quote and #rt), but due to the structure of the network, they appear less popular to users. This occurs either because people who use these hashtags do not have many followers ($\text{Cov}\{f(X), d_o(X)\} < 0$), or the attention of their followers is diluted because they follow many others ($\text{Cov}\{f(U), \mathcal{A}(V)\} < 0$). For example, for #oscars, both of the covariances are negative. Some hashtags also have $B_{\text{local}}$ and $B_{\text{global}}$ with opposite signs, meaning that one measure overestimates the prevalence of the hashtag, while the other underestimates it. Many political hashtags in our sample fall in this category, including #sotu, #occupy, #marriageequality. Additional examples of these hashtags, as well as negatively biased hashtags, are listed in Supplementary Note 2.

**Estimating global prevalence via polling**. The aim of polling is to estimate the global prevalence $\mathbb{E}\{f(X)\}$ of an attribute by sampling individuals and averaging their answers to a specific question. The accuracy of a poll depends on two key factors: (i)

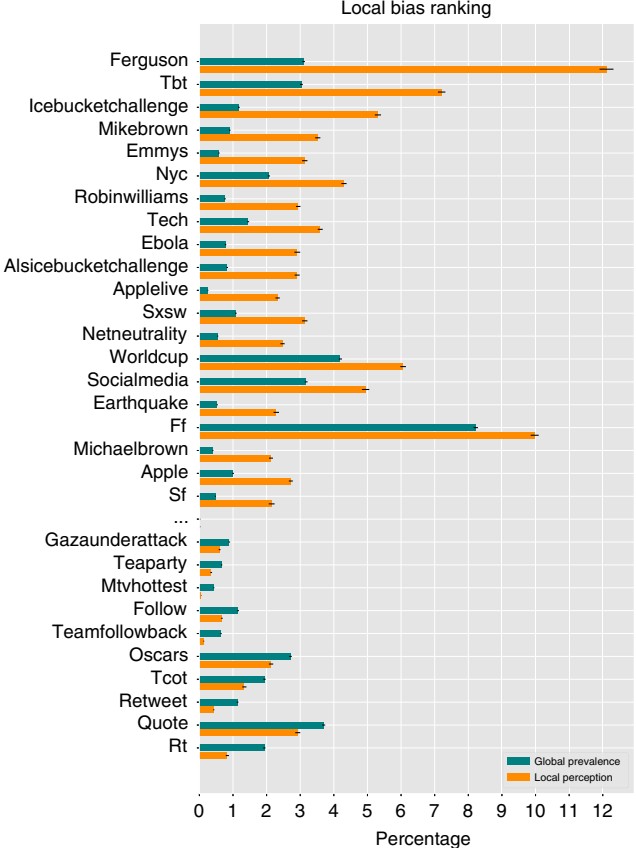

**Fig. 3 The ranking of popular Twitter hashtags based on Local Bias.** Top-20 and bottom-10 are included in the ranking. The bars compare $\mathbb{E}\{f(X)\}$ (global prevalence) and $\mathbb{E}\{q_f(X)\}$ (local perception) and include 95% confidence intervals. The hashtags can appear to be much more popular than they actually are (e.g. #ferguson) or, they can appear to be less popular (e.g. #oscars) due to local perception bias. Definition of some hashtags: #mike(/michael)brown and #ferguson (an 18-year-old African American male killed by police), #tbt (Throwback Thursday—for posting an old picture on Thursdays), #ff (Follow Friday—introducing account worth following), #tcot (Top Conservatives On Twitter), #rt (Retweet).

the method of sampling individuals and (ii) the question asked of them. We propose a practical polling algorithm (see Algorithm 1 in Supplementary Note 3) that differs from the currently used polling algorithms in both aspects. First, our algorithm samples random followers, instead of random individuals, by selecting $b$ individuals from the distribution

$$p_v = \frac{d_i(v)}{\sum_{v' \in V} d_i(v')}, \quad \forall v \in V.$$

Second, the sampled individuals are asked about their perceptions instead of their own attribute: "What do you think is the fraction of individuals with attribute 1?" Their perceptions are then aggregated in a polling estimate:

$$\hat{f}_{\text{FPP}} = \frac{1}{b} \sum_v q_f(v). \quad (14)$$

The key idea behind our follower perception polling (FPP) algorithm is to sample individuals who have more friends, as this allows them to aggregate more information. According to the friendship paradox (Eq. (5)), random followers have, on average, more friends than random individuals do. As a result, the variance of their perceptions will be smaller than that of random individuals, and hence it will result in a more accurate estimate of

the global prevalence of the attribute. We analytically show (see the "Methods" section) that (i) the bias of the estimate $\hat{f}_{\text{FPP}}$ produced by the FPP algorithm is equal to the global perception bias $B_{\text{global}}$ and, (ii) variance of the estimate $\hat{f}_{\text{FPP}}$ is bounded from above by a function of the correlation between out-degree and the attribute, as well as spectral properties of the network given by the second largest eigenvalue of the bibliographic coupling matrix.

The FPP algorithm assumes that every node has a non-zero in-degree and out-degree. To evaluate the performance of this polling algorithm, we extract a subgraph of 5409 Twitter users from our dataset with the same properties. We use the FPP algorithm to estimate the popularity of the 500 most frequent hashtags mentioned by users in this subgraph. We compare the performance of the proposed FPP algorithm on this induced subgraph to two alternative algorithms:

1. Intent polling (IP): asks random users whether they used a hashtag (yellow in Fig. 4).
2. Node perception polling (NPP): asks random users what fraction of their friends used the hashtag (orange in Fig. 4).
3. FPP: asks random followers what fraction of their friends used the hashtag (green in Fig. 4).

NPP differs from IP in terms of the questions asked: random nodes are asked about their perception in NPP, whereas they are asked about their own attribute in IP. FPP differs from NPP in terms of the sampling method: random followers are sampled in FPP, while random node sampling is used in NPP. Hence, comparing the performance of IP with NPP will illustrate the benefit of polling perceptions instead of attributes, and comparing the performance of FPP with NPP will illustrate the benefits of the friendship paradox-based sampling.

Figure 4a shows the bias of estimates produced by polling algorithms for a fixed sampling budget $b = 25$, which corresponds to querying 0.5% of the nodes. As shown in the analysis of the polling algorithm (see the "Methods" section), FPP produces biased estimates for each hashtag (Fig. 4a), given by $B_{\text{global}}$ value for that hashtag, although it produces a smaller variance estimates (Fig. 4b). Hence, in terms of the mean squared error, defined as

$$\text{MSE}\{T\} = \text{Bias}\{T\}^2 + \text{Var}\{T\}$$

for an estimate $T$, FPP estimates are more accurate compared to both IP and NPP for most hashtags (Fig. 4c). Increasing the sampling budget decreases performance gap between FPP and the other two algorithms (Fig. 4d). However, even with $b = 250$ (5% of the nodes polled), FPP outperforms IP in more than 80% of the cases, and it outperforms NPP in more than 55% of the cases.

The variance of the polling estimate (Eq. (19)) is bounded by an expression that includes $\lambda_2$, the second largest eigenvalue of the degree-discounted bibliographic coupling matrix. For the Twitter data $\lambda_2 = 0.5984$. Equation (19) with this value serves as the upper-bound of $\text{Var}(\hat{f}_{\text{FPP}})$ for all 503 hashtags. The bound is quite loose and could be tightened in future work.

*Follower sampling heuristic:* The FPP algorithm assumes that followers are obtained by sampling nodes with probabilities proportional to their in-degree, or equivalently, sampling links at random from the network and then selecting the endpoint of the link. This is feasible when the entire network is known, or when the links have integer IDs, which can be uniformly sampled from a range of IDs. In many cases, neither strategy is feasible, either because the network is too large, or it does not allow access to individually indexed links. In that case, we can use the following heuristic to sample followers: select a node at random and ask her to nominate a random follower. Supplementary Note 3 shows that this heuristic can estimate hashtag prevalence almost as accurately as the exact implementation of the FPP algorithm that

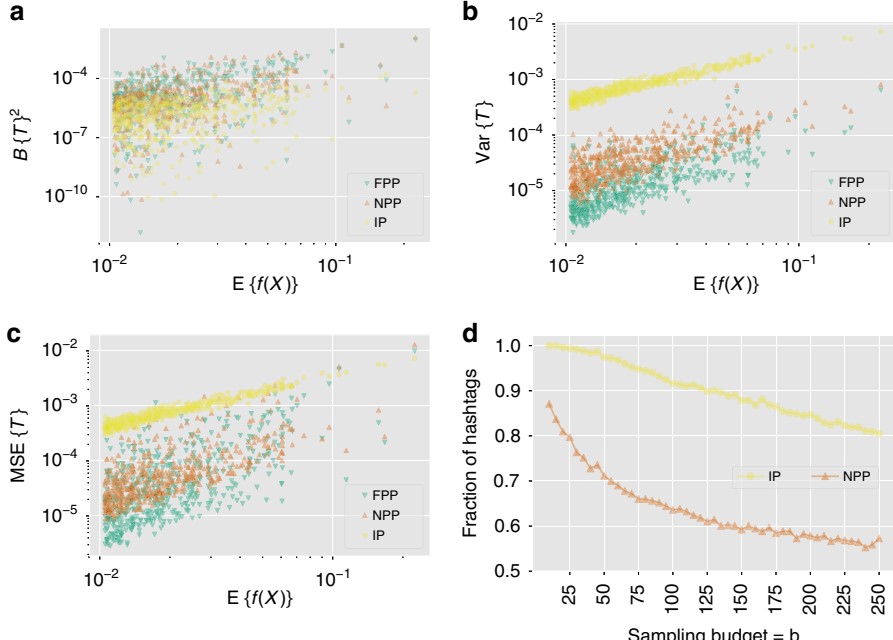

**Fig. 4 Comparison of estimates of the prevalence of Twitter hashtags produced by the polling algorithms.** Variation of **a** squared bias (Bias$\{T\}^2$), **b** variance (Var$\{T\}$), and **c** mean squared error (Bias$\{T\}^2 +$ Var$\{T\}$) of the polling estimate $T$ as a function of a hashtag's global prevalence $\mathbb{E}\{f(X)\}$. Each point represents a different hashtag and a fixed sampling budget $b = 25$. The polling algorithms used are intent polling (IP), node perception polling (NPP) and the proposed follower perception polling (FPP). **d** Fraction of hashtags for which the FPP algorithm outperforms the other two in terms of MSE. The fraction for NPP approaches 0.5, and for IP approaches 0.8 as sampling budget $b$ increases. These figures illustrate that the proposed FPP algorithm achieves a bias-variance trade-off by coupling perception polling with friendship paradox to reduce the mean squared error.

samples nodes proportional to their in-degree. Our intuition for this method is based on the fact that the undirected version of the friendship paradox holds for a random neighbor of a random node as well as a random end of a random link[22].

## Discussion

Social networks can exhibit surprising, even counter-intuitive behaviors. For example, previous work has shown that the "majority illusion" may lead people to observe that the majority of their friends has some attribute, even when it is globally rare[11], and to dramatically underestimate the size of the minority group[6]. These effects arise due to the friendship paradox, which can also bias the observations individuals make in directed networks. Our analysis identifies the conditions under which friendship paradox can distort how popular some attribute or behavior (e.g., drinking, smoking, etc.) is perceived to be, making it appear several times more prevalent than it actually is. The following two conditions amplify perception bias: (1) positive correlation between the attributes of individuals and their popularity (number of followers in a directed network) and (2) positive correlation between the attributes of individuals and the attention of their followers. The first condition suggests that bias exists when popular people have the attribute, for example, engage in risky behavior, have a specific political affiliation, or simply use a particular hashtag. Their influence is amplified when they are followed by good listeners, i.e., people who follow fewer others and thus are able to pay more attention to the influentials. These conditions can be generated by biases in preferences during network formation, driven for example, by homophily[6,23].

We validated these findings empirically using data from the Twitter social network. We measured perceptions of the popularity of hashtags, i.e., words or phrases preceded by a '#' sign that are frequently used to identify topics on Twitter. Such hashtags serve many important functions, from organizing content, to

expressing opinions, to linking topics and people. We measured a hashtag's global prevalence as the fraction of all people using it, and its perceived popularity as the fraction of friends using it. Our analysis identified hashtags that appeared several times more popular than they actually were, due to local perception bias. Such hashtags were associated with social movements, memes, and current events. Interestingly, as our data was collected in 2014, some of the most biased hashtags were #icebucketchallenge and #alsicebucketchallenge, the explosively popular Ice Bucket Challenge. Perception bias could have potentially amplified their spread, as well as the spread of other costly behaviors that require social proof[24]. For example, the #MeToo movement has grown into an international campaign to end sexual harassment and assault in the workplace by highlighting just how endemic the problem is. It spread through online social networks as women posted their own stories of harassment using the hashtag #metoo. Perception bias may have amplified the spread of such hashtags by making them appear more common and thus easier to use.

We also presented an algorithm that leverages friendship paradox in directed networks to estimate the true prevalence of an attribute with smaller mean-squared error than other methods. In essence, the idea behind the algorithm is that perceptions of random followers should have a smaller variance compared to the perceptions of random individuals, because random followers are more informed than random people are, since according to friendship paradox they tend to have more friends. Empirical results confirm that the proposed algorithm outperforms other widely used polling algorithms.

Our work suggests that one way to mitigate perception bias is to alter the local network topology to allow more information to reach the low-attention users. This opens up new research avenues on how link recommendation can alleviate perception bias. However, our empirical study has limitations, namely, the nature of the subsample of the network we considered. Social networks

**Table 1 Properties of the Twitter subgraph.**

| Properties of nodes | | |
|---|---|---|
| Avg. degree | $\overline{d} = E\{d_i(X)\}$ | 123.55 |
| variance of out-degree | $\mathrm{Var}\{d_o(X)\}$ | 30,096.16 |
| variance of in-degree | $\mathrm{Var}\{d_i(X)\}$ | 24,338.66 |
| covariance | $\mathrm{Cov}\{d_i(X), d_o(X)\}$ | 14,226.32 |
| Properties of friends and followers | | |
| Friend's avg. out-degree | $E\{d_o(Y)\}$ | 367.14 |
| Friend's avg. in-degree | $E\{d_i(Y)\}$ | 238.68 |
| Follower's avg. in-degree | $E\{d_i(Z)\}$ | 320.54 |
| Follower's avg. out-degree | $E\{d_o(Z)\}$ | 238.68 |

are huge, necessitating analysis of subgraphs sampled from the entire network. However, by leaving out some nodes, data collection process itself may distort the properties of the sample. Specifically, since we observed only the outgoing links from the seed nodes, we do not have information about the followers of these nodes. Addressing the limitations of analysis imposed by sampling is an important research direction. Despite this limitation, our work shows that friendship paradox can lead to surprising biases, especially in directed networks, and suggests potential strategies for mitigating them.

## Methods
**Data**. The dataset used in this study was collected from Twitter in 2014. We started with a set of 100 users who were active discussing ballot initiatives during the 2012 California election and expanded this set by retrieving the accounts of the individuals they followed and reached a total of 5599 users. We refer to these individuals as seed users. Next, we identified all friends of the seed users, collecting all directed links that start with one of the seed users. We then collected all posts made by the seed users and their friends—over 600K users in total—over the period June–November 2014. The posts include their activity, i.e. tweets and retweets. These tweets mention more than 18M hashtags. With this data-collection approach, seed users are fully observed (their activity and what they see in their social feeds), and their friends are only partially observed (only their activity).

Table 1 reports properties of the Twitter dataset, considering only the seed users. Note that the average degree $\overline{d}$ (where, $\overline{d} = \mathbb{E}\{d_o(X)\} = \mathbb{E}\{d_i(X)\}$) is relatively large at 123.55. However, since the distribution of the in- and out-degree is highly heterogeneous, the variance of the in- and out-degrees is relatively large (two orders of magnitude compared to $\overline{d}$). The covariance between the in- and out-degrees of nodes is also relatively large with a correlation coefficient

$$\rho\{d_i(X), d_o(X)\} = \mathrm{Cov}\{d_i(X), d_o(X)\}/\sqrt{\mathrm{Var}\{d_o(X)\}\mathrm{Var}\{d_i(X)\}} = 0.52.$$

Due to the relatively large variance of the in- and out-degree distributions, the expected out-degree of a random friend ($\mathbb{E}\{d_o(Y)\}$) and the expected in-degree of a random follower ($\mathbb{E}\{d_i(Z)\}$) are larger than the average degree $\overline{d}$ (see Eq. (5)). Note also that, due to positive covariance between the in- and out-degrees of nodes, the expected in-degree of a random friend ($\mathbb{E}\{d_i(Y)\}$) and the expected out-degree of a random follower ($\mathbb{E}\{d_o(Z)\}$) are also larger than $\overline{d}$, as stated in Eq. (7).

**Friendship paradox-based polling: performance analysis**. The accuracy of a poll depends on the method of sampling respondents and the question asked of them. For example, in the case of estimating an election outcome, asking people "Who do you think will win?" (expectation polling) is better than "Who will you vote for?" (IP)[25]. This is because in expectation polling, an individual names the candidate more popular among her friends, thus summarizing a number of individuals in the social network, rather than provide her own voting intention. Our FPP algorithm is motivated by[25–27], which shows that polling methods asking individuals to summarize information in their neighborhood outperform polling methods that ask only about the attribute of each individual. Dasgupta et al.[26] studied the polling problem analytically in the context of an undirected network and, proposed a method to obtain an unbiased estimate of the global prevalence with bounds on its variance. The analysis of the FPP algorithm for directed graphs is motivated by these results in ref. [26] for undirected social networks. Nettasinghe and Krishnamurthy[27] proposed to ask the simple question "What fraction of your neighbors have the attribute 1?" (neighborhood expectation polling) from randomly sampled neighbors (instead of random nodes) on undirected social networks. In this case, sampled individuals will provide the average opinion among their neighbors. Further, since random friends have more friends than random individuals, this approach would yield an estimate with a smaller variance than asking it from

random nodes. Motivated by these works, the FPP algorithm exploits the friendship paradox on directed networks to obtain a statistically efficient estimate of the global prevalence of an attribute using biased perceptions of random followers.

Recall that in order to reduce the variance, the FPP algorithm polls perceptions $q_f(Z)$ of random followers $Z$ instead of attributes $f(X)$ of random individuals $X$. However, it is not guaranteed that the estimate $\hat{f}_{FPP}$ will be unbiased. The following result shows that the bias of the FPP algorithm is the same as the global perception bias $B_{global}$.

The bias of the estimate $\hat{f}_{FPP}$ computed by the FPP algorithm (see Supplementary Note 3 for the derivation) is equal to the global perception bias:

$$\mathrm{Bias}(\hat{f}_{FPP}) = \mathbb{E}\{\hat{f}_{FPP}\} - \mathbb{E}\{f(X)\} = B_{global} \tag{15}$$

Hence, the same factors (specified in Eq. (8)) that increase (decrease) the global perception bias will increase (decrease) the bias of the estimate $\hat{f}_{FPP}$ produced by the FPP algorithm. The aim of the FPP algorithm is to compensate for the bias $B_{global}$ of the algorithm with a reduced variance and thereby achieve a smaller mean squared error. Also, we highlight that FPP algorithm can be modified to generate an unbiased estimate by replacing Eq. (14) with

$$\hat{f}_{FPP}^{Unbiased} = \frac{1}{b}\sum_{v \in S}\frac{1}{Np_v}\sum_{u \in \mathrm{Fr}(v)}\frac{f(u)}{d_o(u)}. \tag{16}$$

The unbiased estimate $\hat{f}_{FPP}^{Unbiased}$ is based on the concept of social sampling proposed in ref. [26] for undirected social networks where, queried individuals provide a weighted value of their friends' attributes in a manner that results in an unbiased estimate. This estimate is useful in contexts where unbiasedness is preferred over mean-squared error to assess the performance of the estimate. However, this does not result in an intuitive and easily implementable algorithm similar to the FPP algorithm, since the modified estimate $\hat{f}_{FPP}^{Unbiased}$ involves each sampled individual calculating a weighted average of the attributes of her neighbors.

Before analyzing the variance of the polling estimate $\hat{f}$ produced by the FPP algorithm, we digress briefly to review the bibliographic coupling matrix. Bibliographic coupling originated from the analysis of citation networks[28], and is used to symmetrize a directed graph by transform it into an undirected graph for purposes of clustering, etc. The bibliographic coupling matrix $B$ of a directed graph with adjacency matrix $A$ is defined as $B = AA^T$. Hence, the weight of the link between nodes $i$, $j$ in the new undirected graph is $B(i, j) = \sum_{v \in V}A(i, v)A(j, v)$ which corresponds to the number of mutual followers of $i$ and $j$. Hence, the weight of the link between two nodes $i$ and $j$ in $B$ is the number of individuals who follow both of these nodes. This conveys the similarity of $i$, $j$ in terms of the number of mutual followers. However, when determining the similarity of two nodes $i$, $j$ using $B$, a mutual follower with a large number of friends (a likely scenario), is weighted the same as a mutual follower with a small number of friends (a rarer scenario). Hence, the latter type of mutual follower should be given more weight compared to the former type when evaluating the similarity of two nodes. Similarly, the number of followers of $i$ and $j$ should also be taken into consideration when assessing their similarity. Based on these observations, Satuluri and Parthasarathy[29] proposed the degree-discounted bibliographic coupling matrix

$$B_d = D_o^{-1/2}AD_i^{-1}A^TD_o^{-1/2} \tag{17}$$

where $D_o$ and $D_i$ are the $N \times N$ dimensional diagonal matrices with $D_o(i, i) = d_o(i)$ and $D_i(i, i) = d_i(i)$, respectively. The $(i, j)$ element of $B_d$ is

$$B_d(i, j) = \frac{1}{\sqrt{d_o(i)d_o(j)}}\sum_{k \in V}\frac{A(i, k)A(j, k)}{d_i(k)}, \tag{18}$$

which discounts the contributions of the nodes $i$, $j$ by their out-degrees and each mutual follower $k$ by her in-degree. Please see refs. [29,30] for more details on the degree-discounted bibliographic coupling.

Returning to the analysis of the estimate $\hat{f}_{FPP}$, we can calculate the upper bound on the variance of this estimate under certain conditions on the structure of the network. Specifically, if the degree-discounted bibliographic coupling matrix $B_d$ is connected, non-bipartite, then

$$\mathrm{Var}(\hat{f}_{FPP}) = \frac{f^T D_o^{1/2}}{bM}(D_o^{-1/2}AD_i^{-1}A^T D_o^{-1/2} - \frac{D_o^{1/2}\mathbb{1}\mathbb{1}^T D_o^{1/2}}{M})D_o^{1/2}f \tag{19}$$

$$\leq \frac{1}{bM}\lambda_2||D_o^{1/2}f||^2 \tag{20}$$

where, $M = \sum_{v \in V}d_i(v)$, $\lambda_2$ is the second largest eigenvalue of $B_d$, $f$ is the $N \times 1$ dimensional vector of binary attributes (see Supplementary Note 3 for the full derivation).

This result shows that the variance of the FPP algorithm depends on the correlation between the out-degrees and attributes $||D_o^{1/2}f||^2$ and the structure of the graph via second largest eigenvalue $\lambda_2$ of the matrix $B_d$. Specifically, a smaller $\lambda_2$ implies that the bibliographic coupling network has a good expansion (i.e. absence of bottlenecks)[31]. Hence, if the nodes in the network $G = (V, E)$ cannot be clustered into distinct groups based on their mutual followers (i.e. bibliographic similarity) then, the variance of the algorithm will be smaller (due to smaller $\lambda_2$).

**Reporting summary**. Further information on research design is available in the Nature Research Reporting Summary linked to this article.

## Data availability

The Twitter network and used hashtags by users data are available in https://osf.io/pjkr9/. Due to Twitter restrictions on sharing raw data, we are unable to share the raw Tweet content.

## Code availability

Codes to generate the results of the paper are available on https://github.com/ninoch/perception_bias.

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

## Acknowledgements

This work was funded in part by Air Force Office of Scientific Research (AFOSR) under Contract No. FA9550-17-1-0327, in part by the Army Research Office (ARO) under Contract No. W911NF-16-1-0306 and by the Defense Advanced Research Projects Agency (DARPA) under Contract No. W911NF-17-C-0094.

## Author contributions

N.A. analyzed data and carried out experiments; B.N. and V.K. carried out theoretical analysis; B.N. completed proofs, K.L., A.A., N.A. and B.N. conceptualized the study; K.L., A.A., N.A. and B.N. wrote the paper; V,K., K.L. and A.A. reviewed the paper.

## Competing interests

The authors declare no competing interests.
