## [Peer Review File · Nature Communications]

Reviewers' comments:

Reviewer #1 (Remarks to the Author):

The authors studied the friendship paradox and its effects on the perception biases both at global and local scales in directed networks, and accordingly analyzed the Twitter dataset containing the information on following-follower relation as well as on the hashtags posted by the Twitter users. They also proposed a new polling algorithm leveraging the friendship paradox. This paper is well-written, relevant, and interesting, whereas I also find several issues to be resolved:

In page 6, the authors defined the global perception bias by comparing the expected attribute of random friends to that of random nodes in Eq. 8, which makes sense. Then what could it mean if the global perception bias was defined using the expected attribute of random followers, i.e., $E\{f(Z)\} - E\{f(X)\}$? Can it be related to the viewpoint of the information spreader rather than the information perceiver?

In page 7, the authors defined the local perception bias, but later they also introduce the individual-level perception bias in page 11. By comparing the equations for those definitions, it seems that the local perception bias is just an expectation value of individual-level perception biases or an average of individual-level perception biases. It would be clearer if the individual-level perception bias is defined earlier than the local perception bias.

In page 8, the authors argue about the relation between the global and local perception biases, by focusing mostly on the case with positive correlations between f and d_o (out-degree) and between f and $1/d_i$ (inverse of in-degree). I am curious other cases, e.g., when f and $1/d_i$ are negatively correlated. This was indeed mentioned in page 9, but can the authors discuss such situations more systematically? For example, if f and $1/d_i$ are negatively correlated, the local perception bias is smaller than the global perception bias. On the top of this, what would happen if f and d_o are also negatively correlated? The statistics of such values for different hashtags can be also helpful to sort out hashtags showing different behaviors in terms of the global and local perception biases. Then the authors can even categorize hashtags using such statistics and relate this observation to the nature of, or contexts of hashtags too. In this direction, there could be a lot more interesting results than in the current paper.

Another comment regarding this issue is as follows: Let us consider the case with a positive correlation between f and d_o , as well as a positive correlation between f and $1/d_i$, leading to a positive correlation between d_o and $1/d_i$ (here d_o is for the friend and d_i is for the follower).

This can tell us something about the degree assortativity (or dissortativity) in the network structure in the directed networks of interest. In this sense, it is remarkable that the effects of the correlation between degree and attribute and the Newman's assortativity measure for degree-degree correlations on the generalized friendship paradox in undirected networks have been systematically studied in Ref. [15].

As the theoretical results and empirical analysis reported in this paper focus on the so-called egocentric network (i.e., an ego and its neighbors/friends/followers) rather than on the global properties of the network, I guess the scope of the paper can be extended to take the global network structure into account. In addition, I wonder how such degree-degree correlation between out-degree of the friend and in-degree of the follower can affect the results of the paper, e.g., the second largest eigenvalue of the degree-discounted bibliographic coupling matrix. It would be also interesting if the value of the upper bound of the variance of the algorithm proposed by the authors is calculated from the Twitter dataset.

Finally, the Twitter users can follow each other (i.e., mutual following). Such mutual following is quite common in online social networking services, and might have a strong impact on the perception bias and information spreading, etc. Is it possible to extend the results in the paper to incorporate such mutual following?

Reviewer #2 (Remarks to the Author):

This is a very interesting paper, and is largely well-written. The major claim is that individuals' perceptions of the global popularity of some topic may be strongly influenced by their local connections, via the friendship paradox/majority illusion. This finding is certainly highly topical, and of interest to the wider field. The result builds upon previous work by the authors, but I suspect would still be quite novel to the community, due to the friendship paradox being not widely known.

I think the analysis that has been done is sound, however I have a few broad questions:

1. The authors present an algorithm for estimating the true popularity of some topic by using the network structure coupled with a cleverly-worded question. However, the number of cases where we know the full network structure (for, say, political polling) is very small, or in cases where we know the network structure (e.g., Twitter as is studied here), we can generally compute the true global prevalence anyway (by counting hashtags). Examining the former case, where we have

incomplete network structure seems very important to consider in order to build the case for why Friendship Perception Polling is useful. This could be done through an experiment where edges in the network are subsampled, for example.

2. There is some discussion of "influence of an interaction" along edges, but this is defined only in terms of node degree, which is a poor proxy for true influence. Defining influence in terms of node degree effectively treats all edges as equal, which is surely not the case. Some edges (such as between family members) will be more influential than others, and these edge weights should be taken into account. Suitable proxies for influence might be rate of messaging between nodes, node similarity, etc.

3. Minor issues/questions:

- the construction of the definitions of global and local perception suggest that it is possible to also find the variance of these measures. Does this make it possible to put confidence intervals on, e.g., the bars in Fig. 3?

- the paper overall would benefit from a good review of the English. For example, the survey question for Follower Perception Polling is missing a "the", and this occurs frequently throughout the manuscript.

- why the scatter in the top right hand panel of Fig. 1? Is this related to Twitter's requirement that users need $n_{\text{followers}} > n_{\text{friends}}$ above a certain limit (I believe it is 2000 or so)? Does this (arbitrary) filtering on the data affect the results?

Response to the Reviewers' Comments

“Friendship Paradox Biases Perceptions in Directed Networks”

Nazanin Alipourfard, Buddhika Nettasinghe, Andres Abeliuk,
Vikram Krishnamurthy, and Kristina Lerman

We wish to thank the referees for their detailed and constructive feedback. Their insights were instrumental to the revision of the manuscript, motivating us to add new analysis of the perception bias highlighting the differences between global and local perception biases, as well as new results pertaining to the scalability of the proposed polling algorithm. Now that the presentation of the paper has been improved with the clarifications and new results, the importance and significance of the work should be much clearer.

REVIEWER 1

- 1.) *In page 6, the authors defined the global perception bias by comparing the expected attribute of random friends to that of random nodes in Eq. 8, which makes sense. Then what could it mean if the global perception bias was defined using the expected attribute of random followers, i.e., $Ef(Z) - Ef(X)$? Can it be related to the viewpoint of the information spreader rather than the information perceiver?*

Response: We thank reviewer for raising this question, which introduces a new variant of perception bias. According to our definition (in Sec. 2), a random friend Y is an individual sampled with a probability proportional to the out-degree and a random follower Z is an individual sampled with a probability proportional the in-degree. As such, a random friend Y can be thought of as a person being observed (or being perceived) whereas a random follower Z is a person who is observing (or perceiving). Our definition of perception bias — $B_{global} = \mathbb{E}\{f(Y)\} - \mathbb{E}\{f(X)\}$ — compares the opinion of a “*random person being observed*” with the global (true) prevalence. The new case reviewer points out — $\mathbb{E}\{f(Z)\} - \mathbb{E}\{f(X)\}$ — compares the opinion of a “*random observer*” with the global prevalence. We have added discussion of this new case to Section 2.2.1.

- 2.) *In page 7, the authors defined the local perception bias, but later they also introduce the individual-level perception bias in page 11. By comparing the equations for those definitions, it seems that the local perception bias is just an expectation value of individual-level perception biases or an average of individual-level perception biases. It would be clearer if the individual-level perception bias is defined earlier than the local perception bias.*

Response: Reviewer is correct in pointing this out. We reorganized Sec. 2.2.2 according to this suggestion.

- 3.) *In page 8, the authors argue about the relation between the global and local perception biases, by focusing mostly on the case with positive correlations between f and d_o (out-degree) and between f and $1/d_i$ (inverse of in-degree). I am curious other cases, e.g.,*

when f and $1/d_i$ are negatively correlated. This was indeed mentioned in page 9, but can the authors discuss such situations more systematically? For example, if f and $1/d_i$ are negatively correlated, the local perception bias is smaller than the global perception bias. On the top of this, what would happen if f and d_o are also negatively correlated? The statistics of such values for different hashtags can be also helpful to sort out hashtags showing different behaviors in terms of the global and local perception biases. Then the authors can even categorize hashtags using such statistics and relate this observation to the nature of, or contexts of hashtags too. In this direction, there could be a lot more interesting results than in the current paper.

Response: We thank reviewer for the suggestion, as it led to new analysis which was quite informative. We added the analysis to a new Section 2.2.3, and summarize it below. We included a new figure showing the spectrum of covariances on which the hashtags fall, which appears in the Supplementary Information Section S7.

Case 1 : $\text{Cov}\{f(U), \mathcal{A}(V)|(U, V) \sim \text{Uniform}(E)\} \geq 0$ and $\text{Cov}\{f(X), d_o(X)\} \geq 0$.

In this case, B_{global} and B_{local} are both positive and local bias is greater than global bias i.e $B_{local} \geq B_{global} \geq 0$.

Case 2 : $\text{Cov}\{f(U), \mathcal{A}(V)|(U, V) \sim \text{Uniform}(E)\} \leq 0$ and $\text{Cov}\{f(X), d_o(X)\} \leq 0$.

In this case, B_{global} and B_{local} are both negative and, local bias is smaller than global bias i.e $B_{local} \leq B_{global} \leq 0$.

Case 3 : $\text{Cov}\{f(U), \mathcal{A}(V)|(U, V) \sim \text{Uniform}(E)\}$ and $\text{Cov}\{f(X), d_o(X)\}$ have opposite signs.

In this case the signs of B_{global} and B_{local} can be different. We make this case more precise with the following results:

- (a) If $B_{global} < 0$, then $B_{local} > 0$ if and only if $\text{Cov}\{f(U), \mathcal{A}(V)|(U, V) \sim \text{Uniform}(E)\} > \frac{|B_{global}|}{d}$
- (b) If $B_{global} > 0$, then $B_{local} < 0$ if and only if $\text{Cov}\{f(U), \mathcal{A}(V)|(U, V) \sim \text{Uniform}(E)\} < \frac{-|B_{global}|}{d}$.

We separate the Twitter hashtags into the above three cases - results illustrated in Fig. 1 (also now added to the Supplementary materials section). The majority of hashtags fall into cases 1 and 2, suggesting that local perception bias is larger than the global perception bias.

Case 1: There are 474 hashtags that fall in this case (shown in green) in Fig. 1. These hashtags are used by popular users who are followed by good listeners. These hashtags include 'ferguson', 'tbt', 'icebucketchallenge', 'mikebrown', 'emmys', 'tech', 'nyc', 'ebola', 'robinwilliams', 'sxsw', 'alsicebucketchallenge', *ldots* All hashtags listed as top 20 in Figure 3 belong to this case except #social_media and #ff.

Case 2: Some of the 187 hashtags falling into this case (shown in red in Fig. 1) include 'rt', 'tcot', 'follow', 'retweet', 'oscar', 'teamfollowback', 'leadfromwithin', 'mtvhottest', 'teaparty', 'shoutout', 'pjnet', *ldots* ; examples of these hashtags are the last-10 hashtags of Figure 3 except #quote.

Case 3: This group has 492 hashtags. The hashtags falling into the left quadrant of Fig. 1 (in orange) include 'sotu', 'occupy', 'marriageequality', 'sandy', ...

Figure 1: Value of normalized $\text{Cov}\{f(U), \mathcal{A}(V) | (U, V) \sim \text{Uniform}(E)\}$ and normalized $\text{Cov}\{f(X), d_o(X)\}$ for all hashtags. The color represents 3 cases.

The hashtags falling into the right quadrant include 'quote', 'quotes', 'win', 'news', 'kindle', 'author', ...

- 4.) *Another comment regarding this issue is as follows: Let us consider the case with a positive correlation between f and d_o , as well as a positive correlation between f and $1/d_i$, leading to a positive correlation between d_o and $1/d_i$ (here d_o is for the friend and d_i is for the follower). This can tell us something about the degree assortativity (or dissortativity) in the network structure in the directed networks of interest. In this sense, it is remarkable that the effects of the correlation between degree and attribute and the Newmans assortativity measure for degree-degree correlations on the generalized friendship paradox in undirected networks have been systematically studied in Ref. [15].*

Response: The relationship between degree assortativity and perception bias in directed networks is more subtle than in undirected networks examined in Ref[15]. To illustrate, consider the special case when the attribute is the out-degree: $f = d_o$. This represents maximal correlation between the attribute $f(X)$ and the out-degree $d_o(X)$. According to Eq. (8), there is always global bias in this case, i.e., $B_{global} \geq 0$. However, it can be shown that $B_{local} \geq 0$ if and only if $\mathbb{E}\{\frac{d_o(U)}{d_i(V)}\} \geq 1$. This special case conveys two important aspects related to the point that the reviewer raised. First, the global perception bias alone does not guarantee a local perception bias (even with the largest possible correlation between f and d_o). Second, the correlation between $d_o(U)$ and $1/d_i(V)$ is associated with local perception bias ($B_{local} \geq 0$) if and only if $\text{Cov}\{d_o(U), 1/d_i(V)\} = \mathbb{E}\{\frac{d_o(U)}{d_i(V)}\} - 1 \geq 0$.

We have largely rewritten the analysis section to clarify these relationships. In addition, we have established a connection to the concept of *inversivity* in undirected networks [1]. Briefly, in an undirected network, inversivity is defined as the correlation between $d(U)$ and $1/d(V)$ where d denotes the degree and (U, V) is a uniformly sampled undirected link. Specifically, [1] show that if inversivity is positive, then the local version of the

friendship paradox is larger than the global version of the friendship paradox. Moreover, they show that inversivity and degree assortativity can differ, even in undirected networks.

Note that we can reproduce the result of Kumar et al. by setting $f = d$ in Eq. 14 in an undirected graph. We have added this interesting connection to the revision.

- 5.) *As the theoretical results and empirical analysis reported in this paper focus on the so-called egocentric network (i.e., an ego and its neighbors/friends/followers) rather than on the global properties of the network, I guess the scope of the paper can be extended to take the global network structure into account. In addition, I wonder how such degree-degree correlation between out-degree of the friend and in-degree of the follower can affect the results of the paper, e.g., the second largest eigenvalue of the degree-discounted bibliographic coupling matrix. It would be also interesting if the value of the upper bound of the variance of the algorithm proposed by the authors is calculated from the Twitter dataset.*

Response: Our analysis accounts for global structure, as it considers expectations of network quantities. For example, we define the perception $q_f(v)$ in Eq. (9) as a local quantity, but all our results are based on its expectation $E\{q_f(X)\}$ (X is a random node), hence they are related to global properties of the network. In fact, $E\{q_f(X)\}$ is the product of two global network properties 1) average degree and 2) expected influence of an interaction along a link (in the paragraph following Eq. (12)). It would be interesting to extend analysis to other global network properties, like degree assortativity in directed networks.

As a first step in this direction, we calculated second largest eigenvalue of the degree-discounted bibliographic coupling matrix for the Twitter graph, as reviewer suggested. We confirmed that it bounds the variance of the polling algorithm. The calculation of the bound and a discussion were added to the polling section.

- 6.) *Finally, the Twitter users can follow each other (i.e., mutual following). Such mutual following is quite common in online social networking services, and might have a strong impact on the perception bias and information spreading, etc. Is it possible to extend the results in the paper to incorporate such mutual following?*

Response: Our analysis does not make any assumptions on reciprocation of follow links, so all the results in the paper hold for the case where mutual following is present. In the extreme case where all edges are mutual, the results presented in the paper would correspond to the friendship paradox in undirected graphs. We added discussion of local perception bias in undirected networks to the new section 2.2.3.

REVIEWER 2

- 1.) *The authors present an algorithm for estimating the true popularity of some topic by using the network structure coupled with a cleverly-worded question. However, the number of cases where we know the full network structure (for, say, political polling) is very small, or in cases where we know the network structure (e.g., Twitter as is studied here), we can generally compete the true global prevalence anyway (by counting hashtags). Examining the former case, where we have incomplete network structure seems very important to consider in order to build the case for why Friendship Perception*

Polling is useful. This could be done through an experiment where edges in the network are subsampled, for example.

Response: Reviewer raises a good point, which we addressed in the revision. We use an alternate sampling heuristic that picks a node at random and then picks a random follower of this node. We confirmed this heuristic produces estimates (of hashtag popularity) that are close to those produced by the exact follower sampling method that relies on knowledge of the full network. We describe the alternate sampling heuristic in the polling algorithm section (see paragraph "Follower sampling heuristic") and added validation figures to Supplementary Information, Section S9.

- 2.) *There is some discussion of "influence of an interaction" along edges, but this is defined only in terms of node degree, which is a poor proxy for true influence. Defining influence in terms of node degree effectively treats all edges as equal, which is surely not the case. Some edges (such as between family members) will be more influential than others, and these edge weights should be taken into account. Suitable proxies for influence might be rate of messaging between nodes, node similarity, etc.*

Response: Extending friendship paradox to weighted directed edges to model the influence of individual friends is indeed an interesting direction for research. We clarified our assumption of uniform influence and also listed this new research question as a potential future direction.

- 3.) Minor issues/questions:

- The construction of the definitions of global and local perception suggest that it is possible to also find the variance of these measures. Does this make it possible to put confidence intervals on, e.g., the bars in Fig. 3?

Response: We added 95% confidence intervals to Figure 3.

- the paper overall would benefit from a good review of the English. For example, the survey question for Follower Perception Polling is missing a "the", and this occurs frequently throughout the manuscript.

Response: We planned to have the manuscript professionally proofread. Unfortunately, the person providing these services at our organization has suffered a stroke recently. Instead, we proofread the manuscript carefully to improve the grammar.

- why the scatter in the top right hand panel of Fig. 1? Is this related to Twitter's requirement that users need #followers > #friends above a certain limit (I believe it is 2000 or so)? Does this (arbitrary) filtering on the data affect the results?

Response: We believe that the noise likely stems from Twitter's follow limits. When individuals reach the limit, they may curate their follow lists more deliberately. However, this curation does not filter links in any way, and should not affect our results. Individuals still receive tweets from their friends in their social feed, no matter how many friends they have. We have added a discussion of this potential source of noise in the discussion of Figure 1.

References

- [1] V. Kumar, D. Krackhardt, and S. Feld, “Network interventions based on inversity: Leveraging the friendship paradox in unknown network structures,” Yale University, Tech. Rep., 2018.

REVIEWERS' COMMENTS:

Reviewer #1 (Remarks to the Author):

The authors revised their manuscript according to all the comments raised in my previous report. The only concern left is that the order of authors in Ref. [17] is wrong.

REVIEWERS' COMMENTS:

Reviewer #1 (Remarks to the Author):

The authors revised their manuscript according to all the comments raised in my previous report. The only concern left is that the order of authors in Ref. [17] is wrong.

- **DONE.**